# Genetic Relatedness of 5-Year Isolates of *Clostridioides difficile* Polymerase Chain Reaction Ribotype 017 Strains in a Hospital

**DOI:** 10.3390/antibiotics10101229

**Published:** 2021-10-09

**Authors:** Jieun Kim, Mi-Ran Seo, Bongyoung Kim, Jinyeong Kim, Mi-Hyun Bae, Hyunjoo Pai

**Affiliations:** 1Department of Internal Medicine, College of Medicine, Hanyang University, Seoul 04763, Korea; quidam76@hanyang.ac.kr (J.K.); sobakas@hanyang.ac.kr (B.K.); 2Molecular Research PCL, Inc., Seoul 08517, Korea; thorn8318@naver.com; 3Department of Internal Medicine, Hanyang University Guri Hospital, Seoul 04763, Korea; wooyajy@gmail.com; 4Department of Laboratory Medicine, College of Medicine, Hanyang University, Seoul 04763, Korea; mhbae@hanyang.ac.kr

**Keywords:** *Clostridioides difficile*, ribotype 017, genetic relatedness, MLVA, antibiotic resistance

## Abstract

The objective of this study was to analyse the genetic relatedness of *Clostridioides difficile* polymerase chain reaction ribotype 017 (RT017) strains from patients with hospital-acquired *C. difficile* infection (HA-CDI) in a hospital with a high RT017 prevalence. From 2009 to 2013, 200 RT017 strains (26.8%) were collected from 745 HA-CDI patient isolates. They comprised 64 MLVA types, and 197 (98.5%) strains were genetically related to 5 clonal complexes (CCs). The largest cluster, CC-A, included 163 isolates of 40 MLVA types. CC-A accounted for 20% of RT017 strains in 2009 and sharply increased to 94.9% in 2010, 94% in 2011, 86.2% in 2012, and 73.5% in 2013. The other 4 CCs included 20 isolates with 7 MLVA types. The resistance rates of antimicrobials were as follows: clindamycin 100%, moxifloxacin 99%, rifaximin 88.5%, and vancomycin 1%. All isolates were susceptible to metronidazole and piperacillin/tazobactam. Comparing antibiotic resistance among CCs, the geometric mean of the minimum inhibitory concentrations of moxifloxacin, vancomycin, and piperacillin/tazobactam were significantly higher for CC-A isolates than for the other CCs. RT017 clones constantly evolved over the 5 years studied with regard to genetic relatedness. The levels of antibiotic resistance may contribute to the persistence of organisms in the institution.

## 1. Introduction

*Clostridioides difficile* is a leading pathogen that frequently causes hospital infections. The polymerase chain reaction (PCR) ribotype 017 (RT017) is a toxin A-negative and toxin B-positive strain that belongs to *C. difficile* clade 4 [1]. RT017 is among several clonal lineages of *C. difficile* known to cause parallel increases in disease severity, mortality, and recurrence [2,3], but has shown similar clinical outcomes to other lineages in some reports [4,5].

In Europe and North America, RT017 strains have been reported to have low prevalence along with common RT027 strains [6]. However, outbreaks of *C. difficile* infections (CDIs) caused by RT017 have been frequently noted in many countries, and a high prevalence of RT017 strains has been reported in Argentina, South Africa, and Asian countries such as Korea and China (especially the areas of Changsha and Fudan) [5,7,8,9,10]. In Korea, RT017 strains have increased in prevalence since 1995 [11], eventually becoming one of the most prevalent PCR RTs; R018 and R017 were the most prevalent RTs followed by R001, R015, R112, R014, R293, R012, R002, R078, R163, R106, and R267 in order of frequency [5]. Although the reason for distinct epidemics of RT017 is not clear, high-level antimicrobial resistance may be one of the key contributing factors.

*C. difficile* can be characterised by various methods, such as restriction endonuclease analysis, North American pulsed-field gel electrophoresis, multilocus sequence typing, multilocus variable-number tandem repeat (VNTR) analysis (MLVA), PCR ribotyping, and whole-gene sequencing [12,13].

In this study, we investigated the prevalence and genetic relatedness of RT017 at a single centre with a high prevalence of RT017 using MLVA and addressed the epidemiologic and genetic relationships among CDIs caused by RT017 strains.

## 2. Results

Over the 5-year period under study, 826 patients were diagnosed with HA-CDI, 745 *C. difficile* isolates were cultivated from stool samples, and toxinotyping and PCR ribotyping were performed. A total of 200 isolates (26.8%) were identified as RT017 strains. All the isolates of RT017 showed toxin A-negative and toxin B-positive toxinotypes (Appendix A).

The proportion of RT017 strains among patients with HA-CDI varied by year (Figure 1).

### 2.1. Clonal Distribution of PCR RT017 Isolates

A total of 64 MLVA types were identified. The VNTR locus CDR4 was constant, except for one isolate. The locus CDR9 was the most variable locus. The diversity of MLVA types decreased by year: 16 types from 20 isolates in 2009, 22 types from 59 isolates in 2010, 19 types from 50 isolates in 2011, 11 types from 29 isolates in 2012, and 12 types from 42 isolates in 2013. A total of 12 common MLVA types comprised 131 of 200 RT017 isolates (65.5%), and they became dominant organisms in 2010 (Appendix A).

The MLVA types of RT017 strains were demonstrated using MST analysis (Figure 2). Of the 200 RT017 isolates, 197 showed genetically related clusters with 5 CCs. In CC-A, 163 (82.7%) strains were included in 40 MVLA types. Ten of the twelve common types were included in CC-A. CC-B was composed of MLVA types 112, 113, and 114, which were observed in 2009 and 2010. Each CC-D and -E was composed of one MLVA type.

According to the department breakdown, isolates from the pulmonary department comprised the largest group with 69 isolates, followed by the nephrology department with 29 isolates (Figure 3). Of the isolates from the pulmonary department, 44 strains (63.8%) were closely related to 1 STRD or less, and 22 strains (75.9%) were closely related to the isolates from the nephrology department.

The MLVA types of the isolates from 2010 and 2011 showed close relatedness, and six MLVA types were shared among them. Isolates from 2012 shared four MLVA types with those from 2013 with small STRD values. However, the association between the isolates from 2011 and 2012 was relatively low (Figure 2); they shared two MLVA types.

### 2.2. Antimicrobial Resistance of RT017 Strains

All RT017 isolates were resistant to clindamycin (CLI) and susceptible to metronidazole (MTZ) and piperacillin/tazobactam (TZP) (Table 1). In the case of vancomycin (VAN), two isolates showed resistance. In contrast, only 2 isolates were susceptible to moxifloxacin (MXF), and 177 isolates (88.5%) were resistant to rifaximin (RFX).

The geometric mean MIC of each antibiotic was determined. Regarding the comparison of MIC values according to isolates from each year, there was no significant difference. The amount of each antibiotic used did not show a statistical difference by year, and there was no significant correlation between the amount of antibiotics used and the geometric mean MIC of each antibiotic (Figure 4).

In terms of MLVA types, the isolates belonging to common MLVA types with more than five isolates showed a significantly higher RFX resistance rate than that of non-common MLVA types (93.9% vs. 78.3%, *p* = 0.002). However, two isolates from VAN-resistant strains belonged to non-common MLVA types.

### 2.3. Comparison of Antimicrobial Resistance among Isolates from CC-A and the Other CCs

The resistance rate of each antimicrobial agent did not show a statistical difference among the isolates of CC-A and the other CCs; the *p* values for MXF, VAN, and RFX were 0.337, 0.337, and 0.775, respectively (Appendix A). However, the geometric mean MIC of MXF was positively correlated with CC-A (rho = 0.46, *p* < 0.001). The values of VAN and TZP also showed positive correlations with CC-A (rho = 0.18, *p* = 0.011 and rho = 0.154, *p* = 0.03, respectively). The geometric mean MICs of CLI, RFX, and MTZ were not correlated with any CC type.

## 3. Discussion

This study analysed the genetic relatedness of *C. difficile* RT017 strains at a single centre with a high prevalence of RT017 using MLVA. The results indicated that the evolution of RT017 continually occurred within the genetic relatedness in endemic situations. One MLVA type (MLVA 78) persisted for 4 consecutive years, and the number of MLVA 78 increased from 1 isolate in 2010 and 2011 to 16 and 15 isolates in 2012 and 2013, respectively. In a report from Taiwan, 40 *C. difficile* RT017 isolates of 10 MLVA types were identified over 5 years, and some MLVA types were observed for 5 consecutive years [14]. In an outbreak in the Netherlands, 41 of 42 isolates (97.6%) were genetically related, and 88% were included in CCs [3]. In our previous study, RT017 infections occurred more frequently in chronically ill patients with higher Charlson’s comorbidity indices and longer durations of hospitalisation than infections by other RTs [5], which indicated that the RT017 infections were primarily hospital infections rather than community infections and persisted longer in the hospital setting.

Antimicrobial resistance is one of the reasons for the endemicity of RT017 strains in hospitals. RT017 strains are consistently resistant to CLI in most studies [3,15]. Susceptibility to rifampin has varied among studies [8,15,16]. In a tuberculosis hospital in South Africa, the rifampin resistance rate of RT017 strains was 99%, and RT017 strains comprised 95.5% of the total *C. difficile* isolates [8]. RT017 strains in a hospital in Hong Kong showed the second-highest resistance to rifampin (58%) [16]. In contrast, the rifampin resistance rate was only 7.7% in a study in North America, where the incidence of RT017 was relatively low [15]. In this study, all RT017 strains were resistant to CLI, and 88.5% of them were resistant to RFX. In our previous study, patients with RT017 infections used 2-fold more CLI or carbapenem and 10-fold more rifampin within the past 2 months than patients with infections by other RTs [5], suggesting a role of rifampin resistance in the spread of RT017 strains in our hospital. Furthermore, the isolates of the dominant CC-A presented the highest geometric mean MICs of MXF, VAN, and TZP, although the resistance rates of these antimicrobials did not differ among the isolates of different CCs. Interestingly, we observed that the isolates of CC-A spread rapidly from 2009 to 2010. These findings strongly suggest that higher antimicrobial resistance contributes to a high prevalence and endemicity of the organism in hospitals.

The fitness of the MLVA 78 type in the hospital setting is not clear. In terms of the antibiotic resistance, the MLVA 78 showed more resistance to MXF than the other MLVA types (data not shown, *p* = 0.042, Mann–Whitney U test). However, MLVA 78 isolates were more susceptible to VAN and TZP (data not shown, *p* < 0.001, *p* = 0.013, Mann–Whitney U test, respectively). TZP was prescribed about two-fold more than VAN or MXF in the hospital, and TZP and VAN use increased in 2012 and 2013 (Figure 4). Therefore, further study including whole-gene sequencing is necessary for the evaluation of fitness of the MLVA 78 type.

The role of environmental contamination via RT017 strains has been clearly documented in a hospital in England [17]. Despite hydrogen peroxide vapour decontamination, the environmental contaminant strains survived. Interestingly, the ancestor of the epidemic RT017 strains was traced back 4 years in the study [17]. Therefore, to reduce RT017 infections in our hospital, active interventions, including contact precautions among patients and environmental cleaning, are necessary.

Although we presented evidence that RT017 strains have constantly evolved over the past 5 years through MLVA, there is still a possibility that there has been a constant influx of new and diverse 017 strains from outside the hospital environment. For example, the community strains which were genetically similar to our strains might be popular in the community. However, as we proposed, RT017 infections occurred more frequently in chronically ill patients with higher Charlson’s comorbidity indices and longer durations of hospitalisation than infections by other RTs [5], and the genetic relatedness of our RT017 isolates from HA-CDI patients was very close, suggesting that the endemic persistence of those strains might be more possible although some RT017 strains influxed into the hospital from the community.

In this study, we presented the genetic relatedness and phenotypic characteristics of all RT017 isolates from a hospital over 5 years. Despite the advantage of the isolates being closely related, the study lacks genetic information; thus, we cannot show the true genetic evolution of important regions such as PaLoc locus. Therefore, it is necessary to study the genetic characteristics of these isolates in the near future.

## 4. Materials and Methods

### 4.1. Patients and Materials

From January 2009 to December 2013, all isolates of *C. difficile* from hospital-acquired *C. difficile* infections (HA-CDI) at Hanyang University Hospital, a 900-bed tertiary care facility in Seoul, Korea, were collected. HA-CDI was confirmed in patients who developed diarrhoea at least 72 h after hospitalisation or within 2 months of their last discharge [18]. According to the results of PCR ribotyping of the isolates, all patients with RT017 strains and isolates were included in this study. The admitted department and date of toxin A and B assays (VIDAS^®^ *C. difficile* Toxin A and B; BioMerieux SA, Marcy l’Etoile, France) tested for each isolate were collected as clinical data via the review of medical records.

The study protocol was approved by the university’s institutional review board (HYG IRB 2018-07-037), and the requirement for written informed consent from patients was waived.

### 4.2. C. difficile Culture and Multiplex PCR for Toxin Gene Detection

After alcohol shock treatment, stool specimens were cultivated on *C. difficile* moxalactam–norfloxacin–taurocholate agar (CDMN-TA agar; Oxoid Ltd., Cambridge, UK) supplemented with 7% horse blood [19]. Colonies of *C. difficile* were identified using Rapid ID 32A (BioMerieux SA, Marcy I’Etoile, France). To identify toxin genes, multiplex PCR was performed using template DNA, as described previously [20]. Template genomic DNA was extracted using an i-genomic BYF DNA Extraction Mini Kit^®^ (iNtRON Biotechnology, Inc., Seongnam-si, Korea). The positive controls were ATCC 43,598 (PCR RT017), ATCC 9689 (PCR RT027), VPI 10,643 (ATCC 43255, PCR RT087), and ATCC 700057, representing A^−^ B^+^ CDT^−^, A^+^ B^+^ CDT^+^, A^+^ B^+^ CDT^−^, and A^−^ B^−^ CDT^−^ RTs, respectively.

### 4.3. PCR Ribotyping

PCR ribotyping was performed using genomic DNA, as described previously [21]. After electrophoresis of the amplified products, the clustering of banding patterns was compared visually with the reference RT017 strain (American Type Culture Collection [ATCC] 43598; Manassas, VA, USA).

### 4.4. MLVA

To trace genetic relatedness, we used MLVA. The six VNTR loci, CDR4, CDR5, CDR9, CDR48, CDR49, and CDR60, were used to perform MLVA [22,23]. The PCR products were sent to Solgent Inc. (Seoul, Korea; http://www.solgent.com/ (accessed on 6 June 2018)) for GENSCAN analysis. Genetic relationships among the genotypes were determined by clustering them according to MLVA type using the number of differing loci and the summed absolute distance as coefficients for calculating the minimum spanning tree (MST). MST analysis was performed using BioNumerics software (version 5.1; Applied Maths NV, Sint-Martens-Latem, Belgium) [24].

When the summed tandem-repeat difference (STRD) was 2 or less and the number of isolates was 2 or more, the cluster was regarded as a clonal complex (CC) [22]. When the STRD was 10 or less, the cluster was regarded as genetically related [25]. MLVA types that included more than five isolates were regarded as common types.

### 4.5. Antimicrobial Susceptibility Testing

Susceptibility tests were conducted with RT017 strains, and the minimum inhibitory concentrations (MICs) of six antimicrobial agents, specifically MTZ, VAN, TZP, CLI, MXF, and RFX, were determined [26]. Brucella agar containing hemin (5 μg/mL), vitamin K1 (10 μg/mL), and 5% horse blood was used for the ASTs, as recommended by the Clinical and Laboratory Standards Institute (CLSI) [27]. The MICs of CLI, MXF, and VAN were measured using the E-test (AB-BIODISK, Solna, Sweden), and those of RFX, TZP, and MTZ were determined using the agar dilution test (Sigma-Aldrich, St. Louis, MO, USA). *C. difficile* ATCC 700,057 was used as a quality control strain for the susceptibility tests.

The resistance breakpoints used were defined as stated by the CLSI and the European Committee on Antimicrobial Susceptibility Testing [27,28].

### 4.6. Amount of Antibiotic Use

The use of six antimicrobial agents in hospitalised patients during the study period was calculated. Rifampin usage was substituted for rifaximin. The total amount of antimicrobial agents was divided by the defined daily dose (DDD) according to the Anatomical Therapeutic Chemical/DDD Index 2018 [29].

### 4.7. Statistical Analysis

IBM SPSS Statistics version 24.0 for Windows (SPSS Inc., Chicago, IL, USA) was used for statistical analysis. Pearson’s chi-square test or Fisher’s exact test was used to analyse categorical variables, and the independent *t*-test or Mann–Whitney U-test was used to analyse continuous variables. Spearman’s rank correlation analysis was performed to assess trends between CCs. Statistical significance was set at *p* < 0.05.

## 5. Conclusions

In summary, RT017 clones have constantly evolved over the 5 years with regard to genetic relatedness. The levels of antibiotic resistance may contribute to the persistence of organisms in the institution.

## Figures and Tables

**Figure 1 antibiotics-10-01229-f001:**
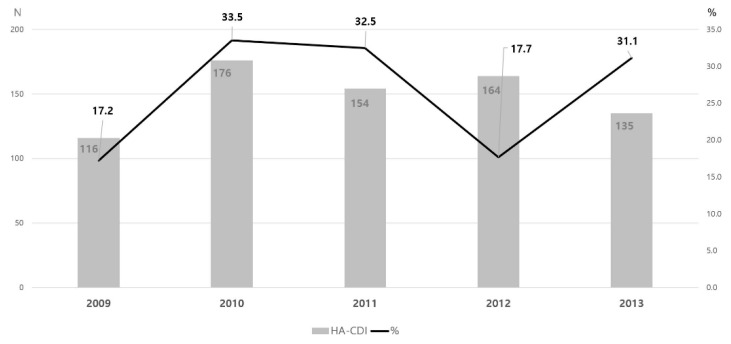
The incidence of healthcare-associated *Clostridioides difficile* infections and proportion of polymerase chain reaction (PCR) ribotype 017 over 5 years. The grey bars represent the number of hospital-acquired *C. difficile* infection (HA-CDI) patients. The black lines represent the proportion of PCR ribotype 017 among HA-CDI patients.

**Figure 2 antibiotics-10-01229-f002:**
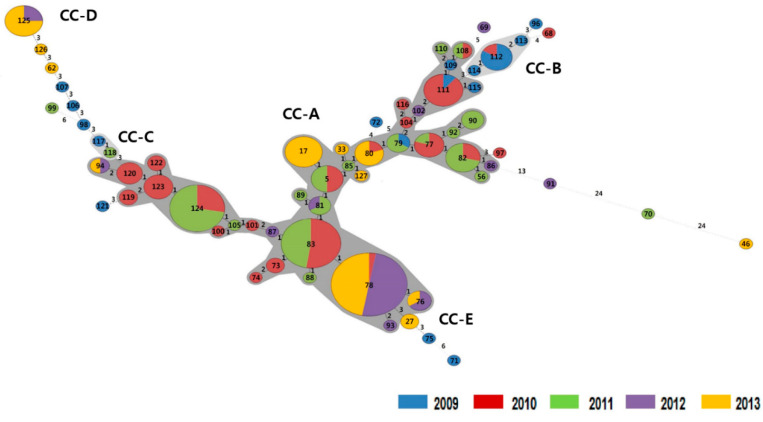
Minimum spanning tree analysis of *Clostridioides difficile* polymerase chain reaction ribotype 017 isolates characterized by multilocus variable-number tandem-repeat analysis (MLVA) over 5 years. Each circle represents a unique MLVA type. The numbers between the circles represent the summed tandem-repeat differences (STRDs) between MLVA types. Clonal complexes (CC-A to CC-E) are marked clusters containing two or more isolates whose MLVA types generated an STRD ≤ 2.

**Figure 3 antibiotics-10-01229-f003:**
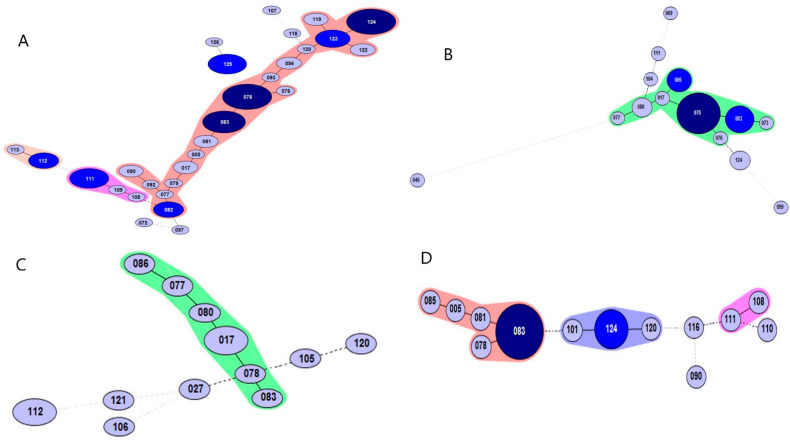
Minimum spanning tree analysis of *Clostridioides difficile* polymerase chain reaction ribotype 017 isolates typed by multilocus variable-number tandem-repeat analysis (MLVA) according to admitted hospital department. Each circle represents a unique MLVA type. (**A**) Pulmonology 66 isolates; (**B**) Nephrology 30 isolates; (**C**) Hemato-oncology 15 isolates; (**D**) Neurosurgery 20 isolates.

**Figure 4 antibiotics-10-01229-f004:**
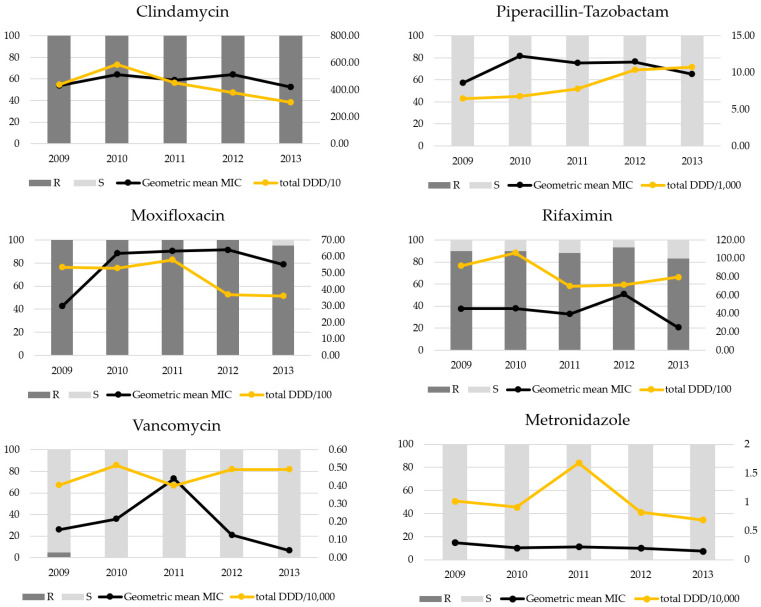
The results of antimicrobial susceptibility tests and geometric mean minimum inhibitory concentrations (MICs) of *Clostridioides difficile* polymerase chain reaction ribotype 017 against each antibiotic over 5 years. Dark grey areas indicate resistance, and light grey areas indicate susceptibility. The black lines indicate geometric mean MICs, and the yellow lines indicate total defined daily dose (DDD), which is divided by 10 to 10,000. R, resistant; S, sensitive.

**Table 1 antibiotics-10-01229-t001:** Antimicrobial susceptibility tests of *Clostridioides difficile* polymerase chain reaction ribotype 017 strains by year.

Antimicrobial Agents	Year (n)	Resistance Rate (%)	MIC (mg/L)		
			MIC_50_	MIC_90_	Geometric Mean
Clindamycin (R ≥ 8 μg/mL)	2009 (20)	100	>256	>256	430.54
	2010 (59)	100	>256	>256	512
	2011 (50)	100	>256	>256	471.14
	2012 (29)	100	>256	>256	512
	2013 (42)	100	>256	>256	420.01
	Total (200)	100	>256	>256	472.77
Moxifloxacin (R ≥ 8 μg/mL)	2009 (20)	100	32	>32	29.86
	2010 (59)	100	>32	>32	61.78
	2011 (50)	100	>32	>32	63.12
	2012 (29)	100	>32	>32	64
	2013 (42)	95.2	>32	>32	55.17
	Total (200)	99	>32	>32	56.69
Vancomycin (R > 2 μg/mL)	2009 (20)	5	0.19	0.5	0.16
	2010 (59)	0	0.38	0.75	0.21
	2011 (50)	2	0.5	1	0.44
	2012 (29)	0	0.19	0.5	0.13
	2013 (42)	0	0.016	0.25	0.04
	Total (200)	1	0.38	0.75	0.16
Piperacillin/Tazobactam (R ≥ 128/4 μg/mL)	2009 (20)	0	16	16	8.57
	2010 (59)	0	16	16	12.21
	2011 (50)	0	16	16	11.31
	2012 (29)	0	16	16	11.45
	2013 (42)	0	16	16	9.75
	Total (200)	0	16	16	10.93
Rifaximin (R ≥ 32 μg/mL)	2009 (20)	90	>64	>64	45.19
	2010 (59)	89.8	>64	>64	45.5
	2011 (50)	88	>64	>64	39.37
	2012 (29)	93.1	>64	>64	61.01
	2013 (42)	83.3	>64	>64	24.54
	Total (200)	88.5	>64	>64	40.19
Metronidazole (R > 2 μg/mL)	2009 (20)	0	0.25	1	0.3
	2010 (59)	0	0.25	0.25	0.2
	2011 (50)	0	0.25	0.25	0.22
	2012 (29)	0	0.25	0.5	0.2
	2013 (42)	0	0.25	0.25	0.15
	Total (200)	0	0.25	0.25	0.2

MIC, minimum inhibitory concentration.

## Data Availability

The data presented in this study are available in Appendix A.

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
