# Peer review of "Genetic Relatedness of 5-Year Isolates of Clostridioides difficile Polymerase Chain Reaction Ribotype 017 Strains in a Hospital"

_antibiotics, 2021, doi:10.3390/antibiotics10101229_

Round 1
Reviewer 1 Report
This is an interesting study looking specifically at the RT017 strain in South Korea. It suggests that not only is RT017 strain prevalent, it is also persistent in the environment of the hospital which may be related to the high resistance to antibiotics, allowing the bacteria to persist longer.
- It may not be in the scope of this paper, but it would be interesting to know, since genetic relatedness was analyzed, if there was evidence of any direct spread from patient to patient in any of these cases.
- In line with item 1, in lines 83-87, the departments in the hospital where the C difficile strains were identified. Were there any clinical characteristics in the patients in these departments that would predispose to C difficile infection? If not, it would strongly suggest that there was direct spread within these wards.
In conclusion, this is a study that adds to the knowledge of RT017 strains and their characteristics and prevalence in Asian countries. The persistence of related strains over the years and the localization of the C difficile strains to particular wards point to a hospital transmission pattern of these infections, which could be stated a bit more concretely. Otherwise, this is a useful study to add to the medical knowledge of C difficile infection.
Author Response
Point 1: It may not be in the scope of this paper, but it would be interesting to know, since genetic relatedness was analyzed, if there was evidence of any direct spread from patient to patient in any of these cases.
Response 1: Because our study was a retrospective study, we could not present a case of direct spread of the organisms among the patients.
Point 2: In line with item 1, in lines 83-87, the departments in the hospital where the C difficile strains were identified. Were there any clinical characteristics in the patients in these departments that would predispose to C difficile infection? If not, it would strongly suggest that there was direct spread within these wards.
Response 2: The patients in department of pulmonology, nephrology, hemato-oncology and neurosurgery were mostly chronic patients who stayed in hospital for a long time. Further, isolates from the patients of each department presented very close relationships in minimum spanning tree analysis.
Reviewer 2 Report
An interesting and important topic, but it should revise thoroughly, some of my specific remarks are listed below:
L22: geometric mean of ……..
L25: antibiotic resistance may not confine the bacteria only in the institution, revise this sentence
L38: Please specify Asian countries and if possible regions also
L39: I also expect CDIs local data in intro section
L191-192, 211: Please italicise bacteria name, double check throughout the paper
L197: toxin A and B assays ordered? Make it clear to your broader readers
L193: Which samples were used to isolate C. difficile and how samples were cultures? How C. difficile was confirmed and maintained in lab for a long time?
L204-205: How DNA was extracted? Was that gDNA? What were modifications, please list in detail
L205: Only for 25 mins? How much volt of current have you supplied? Please add gel image as a supplementary file
L211: I expect to see brief details about culture of C. difficile, saying “described previously” is not enough so, please explain concisely
L213: How RT017 ATCC strain was maintained in lab? Should explain concisely
L225: Please insert relevant references
L237, 238: Readers expect a brief explanation about MIC strip & agar dilution test
L234: Reasons to select those 6 antibiotics?
L238, 241: References are missing
L248: Why no citation? Remove weblink and insert citation
L260-262: Advise to rewrite conclusion
FIG1: I expect exact number/% of each year on the graph
L181-185: This can mention as “study limitation” so, remove it from the discussion section
Author Response
Point 1: L22: geometric mean of …
Response 1: We corrected.
Point 2: L25: antibiotic resistance may not confine the bacteria only in the institution, revise this sentence
Response 2: We understand your opinion. However, we would like to emphasize the point that the antibiotic resistance contributes the persistence of the organisms in an institution for many years.
Point 3: L38: Please specify Asian countries and if possible regions also
Response 3: We corrected: line 37 “a high prevalence of RT017 strains has been reported in Argentina, South Africa, and Asian countries such as Korea and China (especially, Changsha and Fudan area)”
Point 4: L39: I also expect CDIs local data in intro section
Response 4: In our previous article referred in the text (5), the local data is described; “Other ribotypes identified in this study except R017 and R018 were R001, R015, R112, R014, R293, R012, R002, R078, R163, R106, R267 in order of frequency, and 131 strains were undefined ribotypes.”
Point 5: L191-192, 211: Please italicise bacteria name, double check throughout the paper
Response 5: We corrected.
Point 6: L197: toxin A and B assays ordered? Make it clear to your broader readers
Response 6: We changed “ordered” to “tested”.
Point 7: L193: Which samples were used to isolate C. difficile and how samples were cultures? How C. difficile was confirmed and maintained in lab for a long time?
Response 7: We added the sentence at method section; “Stool specimens were grown anaerobically on C. difficile Moxalactam-Norfloxacin-Taurocholate agar (CDMN-TA, Oxoid Ltd., Cambridge, UK) supplemented with 7% horse blood after alcohol shock (19). Colonies of C. difficile were identified by Rapid ID 32A (BioMerieux SA, Marcy l’Etoile, France).”
To make a stock sample of the subcultured strain, colonies grown in CDMN-TA medium were dissolved in a liquid medium containing 25% glycerol and 5% horse blood in thioglycollate medium (Oxoid, CM0173) and stored in a cryogenic freezer at -70°C. ; This sentence is not included in the manuscript.
Point 8: L204-205: How DNA was extracted? Was that gDNA? What were modifications, please list in detail
Response 8: We add the method for DNA extraction at the method section.
Point 9: Only for 25 mins? How much volt of current have you supplied? Please add gel image as a supplementary file
Response 9: The resolving gel of multiplex PCR was run at 100 volt.
This is the figure for multiplex PCR results. Strain number 1,2,5, and 9 showed tcdA/B positive strain. Strain number 4 and 7 showed only 16S rDNA which means non-toxigenic strains. Strain number 3, 6, and 8 showed cdtA/B positive strain as well as tcdA/B positive, which means binary toxin strain. P, positive control (BI/NAP1/027 strain); N, negative control.
Point 10: L211: I expect to see brief details about culture of C. difficile, saying “described previously” is not enough so, please explain concisely
Response 10: We changed the subtitle from “multiplex PCR for toxin gene detection” to “C. difficile culture and multiplex PCR for toxin gene detection”. Further, we added the sentence at line 216; “Stool specimens were grown anaerobically on C. difficile Moxalactam-Norfloxacin-Taurocholate agar (CDMN-TA, Oxoid Ltd., Cambridge, UK) supplemented with 7% horse blood after alcohol shock (19). Colonies of C. difficile were identified by Rapid ID 32A (BioMerieux SA, Marcy l’Etoile, France).”
Point 11: L213: How RT017 ATCC strain was maintained in lab? Should explain concisely.
Response 11: After purchase of RT017 ATCC strain, we had cultured the strain on multiple plates and made and stored the multiple frozen vials of RT017 strain at -70℃. Further, we made multiple vials of DNA extracted for easy use, and froze them.
Point 12: L225: Please insert relevant references
Response 12: We added reference.
Point 13: L237, 238: Readers expect a brief explanation about MIC strip & agar dilution test
Response 13: Because we did not modify the method of MIC strip and agar dilution, we added the reference.
Point 14: L234: Reasons to select those 6 antibiotics?
Response 14: Metronidazole and vancomycin are the antibiotics for the treatment of CDI. Moxifloxacin is the antibiotics which probably caused the outbreak of RT027 in western countries, which might have a strong selection power. Rifaximin is the medication which are used for CDI in European countries. Piperacillin-tazobactam is the antibiotic which might reduce the occurrence of CDI in several papers, because it has a powerful inhibition effect on C. difficile. In near future, we would add fidaxomicin and ridinilazole.
Point 15: L238, 241: References are missing
Response 15: We edited reference.
Point 16: L248: Why no citation? Remove weblink and insert citation
Response 16: We edited reference.
Point 17: L260-262: Advise to rewrite conclusion
Response 17: We changed as “In summary, RT017 clones have constantly evolved over the 5 years with regard to genetic relatedness. The levels of antibiotic resistance may contribute to the persistence of organisms in the institution.”
Point 18: FIG1: I expect exact number/% of each year on the graph
Response 18: According to your recommendation, we edited the figure 1.
Point 19: L181-185: This can mention as “study limitation” so, remove it from the discussion section
Response 19: Limitation of the study is generally written in discussion, and we did not remove the sentence. I hope you understand.
Reviewer 3 Report
This article describes and analyzes screening studies of the presence of certain C. difficile strains in a particular clinic over a five-year observation period (2009-2013). The manuscript is well-formed, without serious remarks, both in form and in content. The topic of antibiotic resistance is currently extremely important, as strains of microorganisms resistant to a wide range of antimicrobial substances are becoming more widespread. Particular attention is drawn to bacteria that can cause so-called hospital infections. It is important to imagine the situation as objectively as possible in order to develop strategies to combat such microorganisms, which is what this material is aimed at, in particular. I believe that it can be published after minor corrections.
Line 94 - Misunderstandings may arise here. If the data refer to Figure 3, it will seem that Figure 3 lacks a legend with colored years, as in Figure 2. If this line discusses the results related to the second figure, then it would be better to indicate this immediately in this line rather than at the end of the paragraph.
Line 101 - Even if such abbreviations for antibiotic names are widely accepted in the scientific and medical communities, authors should be given a transcript when they first mention them.
Line 112 - I don't quite understand why the authors removed this data in the Supplementary section. Perhaps the manuscript would have looked better if this figure had been added to the main body of the article.
Lines 119-125 - Requires data link, such as a picture or table.
Line 163 - If the authors move this picture into the main body of the article, the link will need to be changed here.
Line 200 - Sorry, I could not find and verify this HYG IRB 2018-07-037 number over the Internet. If the authors provided a link to the published protocol, or if this is an internal document of the organization that cannot be published in the public domain - at least to its number on the official website of the clinic, it would be good. As a last resort, one could submit an official paper from the organization, which confirms that this number is indeed associated with this study.
Lines 211-212 - it would be nice to give a link to a specific method of DNA extraction, or describe it here. There are variations in the methods, and our task is to describe the methods as accurately as possible so that the experiment could be repeated easily, without resorting to a literature search.
Authors should also correct minor omissions throughout the text, such as slanting of species names of microorganisms and genes, typos, etc.
Author Response
Point 1: Line 94 - Misunderstandings may arise here. If the data refer to Figure 3, it will seem that Figure 3 lacks a legend with colored years, as in Figure 2. If this line discusses the results related to the second figure, then it would be better to indicate this immediately in this line rather than at the end of the paragraph.
Response 1: Thank you for your comments. We did not mark the year for the Figure 2, because the number of the isolates from each department was not enough for marking the years.
Point 2: Line 101 - Even if such abbreviations for antibiotic names are widely accepted in the scientific and medical communities, authors should be given a transcript when they first mention them.
Response 2: Thank you for your recommendation. We add the abbreviation
Point 3: Line 112 - I don't quite understand why the authors removed this data in the Supplementary section. Perhaps the manuscript would have looked better if this figure had been added to the main body of the article.
Response 3: We change the supplementary figure 1 as figure 4 which located in the main body of the article.
Point 4: Lines 119-125 - Requires data link, such as a picture or table.
Response 4: We add supplementary table 2.
Point 5: Line 163 - If the authors move this picture into the main body of the article, the link will need to be changed here.
Response 5: According to your recommendation, we change the supplementary figure 1 as figure 4 which located in the main body of the article.
Point 6: Line 200 - Sorry, I could not find and verify this HYG IRB 2018-07-037 number over the Internet. If the authors provided a link to the published protocol, or if this is an internal document of the organization that cannot be published in the public domain - at least to its number on the official website of the clinic, it would be good. As a last resort, one could submit an official paper from the organization, which confirms that this number is indeed associated with this study.
Response 6: Thank you for your recommendation. We attached IRB-approved file.
Point 7: Lines 211-212 - it would be nice to give a link to a specific method of DNA extraction, or describe it here. There are variations in the methods, and our task is to describe the methods as accurately as possible so that the experiment could be repeated easily, without resorting to a literature search.
Response 7: We add the method for DNA extraction.
Point 8: Authors should also correct minor omissions throughout the text, such as slanting of species names of microorganisms and genes, typos, etc.
Response 8: Thank you for your recommendation. We correct errors.
Round 2
Reviewer 2 Report
Dear authors,
Thanks so much for submitting the revised version of paper, you have improved a few points, you responded to all points or answered me as a reviewer; but 80% only vaguely to me and not at all translated into the manuscript. You should not only answer the question but also improve the manuscript so, kindly requested to replicate those points on the revised manuscript. I cannot go through all the individual points I have mentioned before; you still have my original list; it would be nice if you could add E-test result photograph, conclusion is same as previous, readers wish to see local data on present paper rather searching in old paper so, it is good to mention concisely taking that paper as a reference.
Author Response
Point 4: L39: I also expect CDIs local data in intro section
Response 4: We added the sentence at line 39: In Korea, RT017 strains have increased since 1995 [11], eventually becoming one of the most prevalent PCR RTs; R018 and R017 were the most prevalent RTs followed by R001, R015, R112, R014, R293, R012, R002, R078, R163, R106, R267 in order of frequency [5].
Point 9: Only for 25 mins? How much volt of current have you supplied? Please add gel image as a supplementary file
Response 9: The resolving gel of multiplex PCR was run at 100 volt.
We added the gel image of multiplex PCR at supplementary figure 1 at line 56.
Point 17: L260-262: Advise to rewrite conclusion
Response 17: The conclusion at line 260 is same as the conclusion in the abstract. “In summary, RT017 clones have constantly evolved over the 5 years with regard to genetic relatedness. The levels of antibiotic resistance may contribute to the persistence of organisms in the institution.”
We do not understand your comment clearly and we are not sure how we rewrite the conclusion. If it is absolutely necessary, please give your advice how we rewrite the conclusion.
